# CSD-AFNet: Computationally Efficient Atrial Fibrillation Classification from ECGs using 2D Causal Strided Dilated Convolutions

Lennert Bontinck, Aranka Steyaert, Hongbing Chen, Tom Dhaene and Dirk Deschrijver

*IDLab*

*Ghent University - imec*

Ghent, Belgium

{Lennert.Bontinck, Aranka.Steyaert, Hongbing.Chen, Tom.Dhaene, Dirk.Deschrijver}@ugent.be

*Abstract*—
**Automated analysis of electrocardiogram (ECG) signals using deep learning (DL) methods has shown substantial promise in atrial fibrillation (AFib) classification, particularly for detecting subtle indicators during normal sinus rhythm and for predicting new-onset AFib. However, many existing state-of-the-art models exhibit high computational demands, characterised by large parameter and floating-point operations (FLOPs) counts. This presents a high barrier to entry for training in budget-limited institutes and hinders the models' deployment on medical edge devices. This paper introduces CSD-AFNet, a computationally efficient DL model specifically designed for AFib-related ECG classification tasks. CSD-AFNet achieves substantial reductions in both parameter and FLOPs counts by replacing expensive temporal convolutions with novel Feature-Preserving Pooled Convolutions (FPP-Convs). FPP-Convs enable the combination of striding and dilation without input feature loss, preserving temporal coverage while reducing the computational cost. The model further incorporates two-dimensional causal padding to prevent temporal leakage in downstream representations. Evaluation on the public CODE-15% and PTB-XL datasets demonstrates that CSD-AFNet matches the classification performance of leading benchmark models while reducing parameter count by a factor of 71 and FLOPs by a factor of 122 compared to the ResNet-10 inspired baseline. These findings support the suitability of CSD-AFNet for practical clinical scenarios, enabling training under resource constraints and efficient inference on medical edge devices, thereby facilitating scalable and cost-effective ECG-based AFib screening and monitoring.**

*Index Terms*—**Electrocardiography, Atrial Fibrillation, Deep Learning, Computational Efficiency, Compact Model, Causal Convolution, Strided Dilated Convolution, Arrhythmia Detection**

## I. Introduction

The electrocardiogram (ECG) is a standard clinical diagnostic tool, valued for its affordability, non-invasive nature, and ease of data acquisition. A conventional 12-lead ECG provides detailed cardiac electrophysiological information, supporting early detection and effective management of cardiovascular conditions.

Funding: This work was supported by the Interreg France-Wallonie-Vlaanderen (FWVL) program and the province Oost-Vlaanderen as part of the VasculAI project and by the Flemish Government via the AI Research Program.

Automated analysis of ECG signals using artificial intelligence, and in particular deep learning (DL), has gained traction due to its ability to deliver rapid, consistent, and accurate diagnostic interpretations. These DL models can surpass the limitations of manual interpretation, which relies heavily on expert pattern recognition.

The application of DL-based ECG analysis to atrial fibrillation (AFib), one of the most prevalent cardiac arrhythmias with substantial clinical impact, demonstrates this potential for improved diagnostic insight. In contrast to traditional methods and clinician-led assessments, DL models can detect subtle patterns indicative of AFib during normal sinus rhythm (NSR) or even before new-onset AFib (Prediction), enhancing both monitoring and risk stratification possibilities [1]–[4]. Furthermore, these models' capacity to analyse large ECG datasets enables retrospective analysis and the development of data-driven screening strategies.

Despite the demonstrated effectiveness of DL models in ECG analysis, many current state-of-the-art (SOTA) architectures are computationally demanding, characterised by large parameter and floating-point operations (FLOPs) counts. During training, models with high parameter and FLOPs counts require substantial graphics processing unit (GPU) memory and processing capacity, significantly increasing training time and cost. This creates a practical barrier for institutions in developing countries or smaller clinical centres, where budget constraints and limited access to high-end GPU infrastructure hinder the ability to train models on locally available data. In deployment settings, particularly on medical edge devices such as the ECG acquisition device itself, the memory requirements associated with large parameter counts and the compute intensity driven by high FLOPs introduce latency and energy consumption challenges. Such constraints make high-complexity models unsuitable for low-power or real-time applications where inference must be fast, efficient, and reliable without access to dedicated GPU resources [5]. Therefore, reducing both parameter and FLOPs count is critical to improving the accessibility, scalability, and practical deployment of DL-based ECG analysis models across a wider range of clinical and resource-constrained environments [3].

Emerging solutions, such as photonics-based neural networks, offer substantial efficiency improvements on a hardware level but necessitate fixed model architectures optimised for these hardware constraints [6]. Such photonic hardware would benefit from compact model designs with minimal parameter and FLOPs count as they require less complex chips to be created.

Recent work, including the ECGencode feature encoder designed for ECG signals, has demonstrated that substantial reductions in computational load are possible without compromising diagnostic accuracy. The ECGencode model 1 by Bontinck et al. [3] achieves classification performance on par with leading models, yet retains a relatively high FLOPs count. In contrast, AFibri-Net by Phukan et al. [5] achieves lower FLOPs and has been shown to run on low-resource inference devices, though at the cost of lower classification performance [3].

To further bridge this gap, and achieve AFibri-Net level efficiency without the performance trade-off, this paper proposes a novel compact DL model specifically developed for ECG-based AFib classification. The proposed CSD-AFNet model integrates strided dilated convolutions to reduce the computational burden of early temporal operations. To solve the issue of missed input features present in the conventional combination of striding and dilation (see Fig. 2), Feature-Preserving Pooled Convolutions (FPP-Convs) are introduced. Furthermore, ECG-specific 2D causal convolutions are employed to eliminate "future leakage", thereby improving the interpretability and fidelity of the latent space representations.

The key contributions of this work are threefold: (1) incorporation of 2D causal padding to ensure temporal consistency in latent 2D ECG representations; (2) introduction of FPP-Convs, enabling strided dilated convolutions without feature loss and with greatly reduced computational cost; and (3) empirical demonstration that CSD-AFNet matches current SOTA AFib classification performance across datasets and tasks, while significantly improving computational efficiency.

## II. BACKGROUND AND RELATED WORK

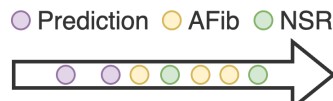

Fig. 1: Temporal distribution of ECGs from AFib-positive patients, categorised as Prediction (pre first AFib), AFib (active episode), and NSR (post AFib without rhythm disturbance).

Recent years have seen substantial progress in the application of DL techniques to AFib-related ECG analysis. While traditional approaches focus on detecting active AFib episodes, a task routinely handled by cardiologists and feasible through algorithmic integration into medical edge devices (i.e., ECG acquisition devices), DL models have extended this scope significantly.

Beyond active rhythm detection, DL-based approaches now tackle more complex predictive tasks such as identifying AFib during periods of NSR and predicting future onset AFib. These tasks cannot be accomplished using conventional diagnostic methods. The three subtypes of AFib-related signals, namely prediction, active AFib, and NSR, from an AFib-positive patient are illustrated in Fig. 1.

Given the growing demand for both prospective and retrospective analysis on resource-limited platforms, computational efficiency has become a central focus. This work introduces CSD-AFNet, a novel DL model that incorporates causal strided dilated convolutions and key concepts from Temporal Convolutional Networks (TCNs) [7], thereby enhancing computational efficiency while preserving diagnostic accuracy.

### A. Computational Efficiency in ECG-based AFib Models

The model proposed by Attia et al. [1], from now on referred to as the Attia model, represents a foundational approach to AFib-related ECG analysis. Their use of a Residual Network (ResNet) with ten residual blocks demonstrated that subtle AFib markers could be detected during NSR, benefitting from skip connections to mitigate potential negative effects of increased model depth [8]. This architecture has since become a common baseline for various AFib and general ECG classification tasks. However, the depth and complexity of ResNets inherently result in high parameter and FLOPs counts, increasing training cost and limiting deployment on edge devices and in resource-constrained settings.

To address these limitations, the AFibri-Net model was proposed by Phukan et al. [5] for AFib classification from raw ECG signals with reduced computational cost. AFibri-Net achieves notable FLOPs efficiency, with demonstrated feasibility for inference on low-resource hardware. However, its high parameter count raises concerns about memory usage and potential overfitting, despite the mitigating effect of the double-descent phenomenon [9].

Additionally, empirical results from the ECGencode study by Bontinck et al. [3] suggest that AFibri-Net's compactness compromises its ability to achieve SOTA classification performance. To strike a better balance between computational cost and diagnostic accuracy, the authors of the ECGencode study introduced ECGencode Model 1, from now on referred to as the ECGencode model, which reduces high-dimensional input to a compact latent space and achieves classification performance comparable to Attia's model for various AFib-related tasks, with orders of magnitude fewer parameters and FLOPs. Despite these benefits, ECGencode still exhibits higher FLOPs than AFibri-Net, primarily due to its use of variable-length kernels in the temporal convolutional layer.

This work addresses both the computational inefficiency and a temporal ambiguity present in ECGencode by introducing enhanced versions of existing convolutional mechanisms: strided dilated convolutions and causal padding.

### B. Strided Dilated Convolution

Strided convolutions reduce the number of kernel applications by increasing the step size, effectively decreasing feature

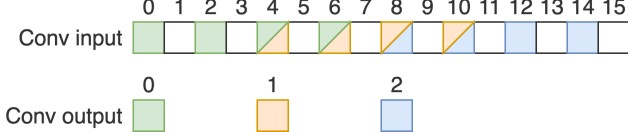

Fig. 2: Visualisation of skipped feature issues when combining striding and dilation in temporal convolutions. The example shows a conventional kernel with size $k = 4$, dilation $d = 2$, and stride $s = 4$ which, given $s$ is a multiple of $d$, results in $\frac{d}{s} = \frac{1}{2}$ of the input features being skipped.

map resolution and overall FLOPs. Conversely, dilated convolutions increase the receptive field by spacing kernel elements, capturing longer temporal dependencies without increasing the kernel size and thus parameter and FLOPs count.

Combining these two operations, however, can result in skipped input features due to compounded downsampling. This issue is visualised in Fig. 2 and is a known limitation.

To avoid skipping input features, a novel Feature-Preserving Pooled Convolution (FPP-Conv) is introduced and used in the proposed CSD-AFNet model, as further discussed in Section III-B. This means CSD-AFNet can benefit from a large temporal reach while maintaining computational efficiency.

### C. Causal Padding in Causal Convolutions

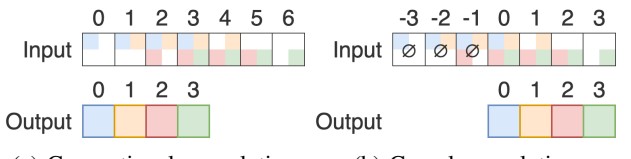

(a) Conventional convolution     (b) Causal convolution

Fig. 3: Illustration of 1D causal padding to prevent "future leakage". (a) In conventional convolutions, outputs include future time steps. (b) Causal convolutions left-pad the input by $(k-1){\cdot}d$ zeros, with $k$ the kernel size and $d$ the dilation rate, restricting outputs to present and past inputs. Examples shown use $k{=}4$, $d{=}1$ with the input row showing the time steps of influence for the single same-coloured output time step.

Causal padding as used in causal convolutions ensures that each convolution output at a given time step is only influenced by present and past inputs, thus avoiding "future leakage" (see Fig. 3). This property is especially important in sequential modelling tasks and is widely employed in domains such as audio signal processing [10].

Causal padding is typically applied to one-dimensional time series data, where an ECG signal could be seen as a 1D time series signal with multiple channels (leads) [11]. However, ECG signals are often represented as 2D matrices to explicitly retain a leads × time shape throughout the model, aiding latent space interpretability. To further aid this intuitive understanding of the latent space, and ensure timepoints don't exhibit the "future leakage" problem, the proposed CSD-AFNet model adopts the causal padding technique for 2D ECG representations as further discussed in Section III-A.

### D. Comparison with Temporal Convolutional Networks (TCN)

Traditional TCNs are end-to-end architectures that model long-range dependencies using stacked causal and dilated convolutions, and have shown promise in ECG analysis [7], [11]. In contrast, CSD-AFNet focuses on efficient temporal feature extraction within a fixed receptive field, using 2D causal padding and FPP-Convs to balance efficiency and interpretability without aiming for global sequence modelling.

### III. CSD-AFNet AFib Classification Model

The proposed CSD-AFNet model, visualised in Fig. 4, builds upon the ECGencode model. Unlike the original ECGencode model that employs variable-length convolution kernels and striding in the Temporal Convolution component, CSD-AFNet uses novel FPP-Convs with a fixed kernel size of 16, stride of 8 and varying dilation rates. This preserves a temporal coverage of 0.03, 0.1, 0.5, and 2 seconds while substantially reducing computational complexity in terms of parameter and FLOPs count. The FPP-Convs are needed to address the issue of skipping features when combining striding and dilation in regular convolutions as discussed in Section II-B and shown in Fig. 2. Section III-B discusses the novel FPP-Convs in more detail.

Another important change compared to the ECGencode model is the use of 2D causal convolutions to preserve temporal causality and eliminate "future leakage" as discussed in Section II-C and shown in Fig. 3. This is discussed in more detail in Section III-A.

### A. Causal Padding for 2D ECG Signals

1D Causal padding as discussed in Section II-C and visualised in Fig. 3 can easily be adopted for 2D ECG signals by left-padding the temporal-axis. The offset is determined by the temporal reach, requiring left-padding of $(k{-}1){\cdot}d$ with $k$ the kernel size and $d$ the dilation rate.

### B. FPP-Conv: Strided Dilated Convolution Without Skipping Features

Combining striding and dilation in convolutional layers often results in incomplete coverage of temporal features, as discussed in Section II-B. To mitigate this, Feature-Preserving Pooled Convolutions (FPP-Convs) are introduced. A FPP-Conv is a two-step process taking a target temporal dilation rate $d$ and target temporal striding rate $s$ together with the desired number of output channels $c$.

First, pooling is applied using both temporal kernel and temporal stride sizes set to $p = min(d, s)$. This results in a temporal-axis downsampling with the new temporal length being $\lceil \frac{n}{p} \rceil$ where $n$ is the original temporal length. Given the pooling operation, a residual stride $s_{res} = \lceil \frac{s}{p} \rceil$ and a residual dilation $d_{res} = \lceil \frac{d}{p} \rceil$ remain, one of which is now equal to 1. Second, a regular convolution is applied using the remaining stride or dilation (whichever exceeds 1).

This two-step process means FPP-Conv retains the target receptive field and desired computational benefits of combining

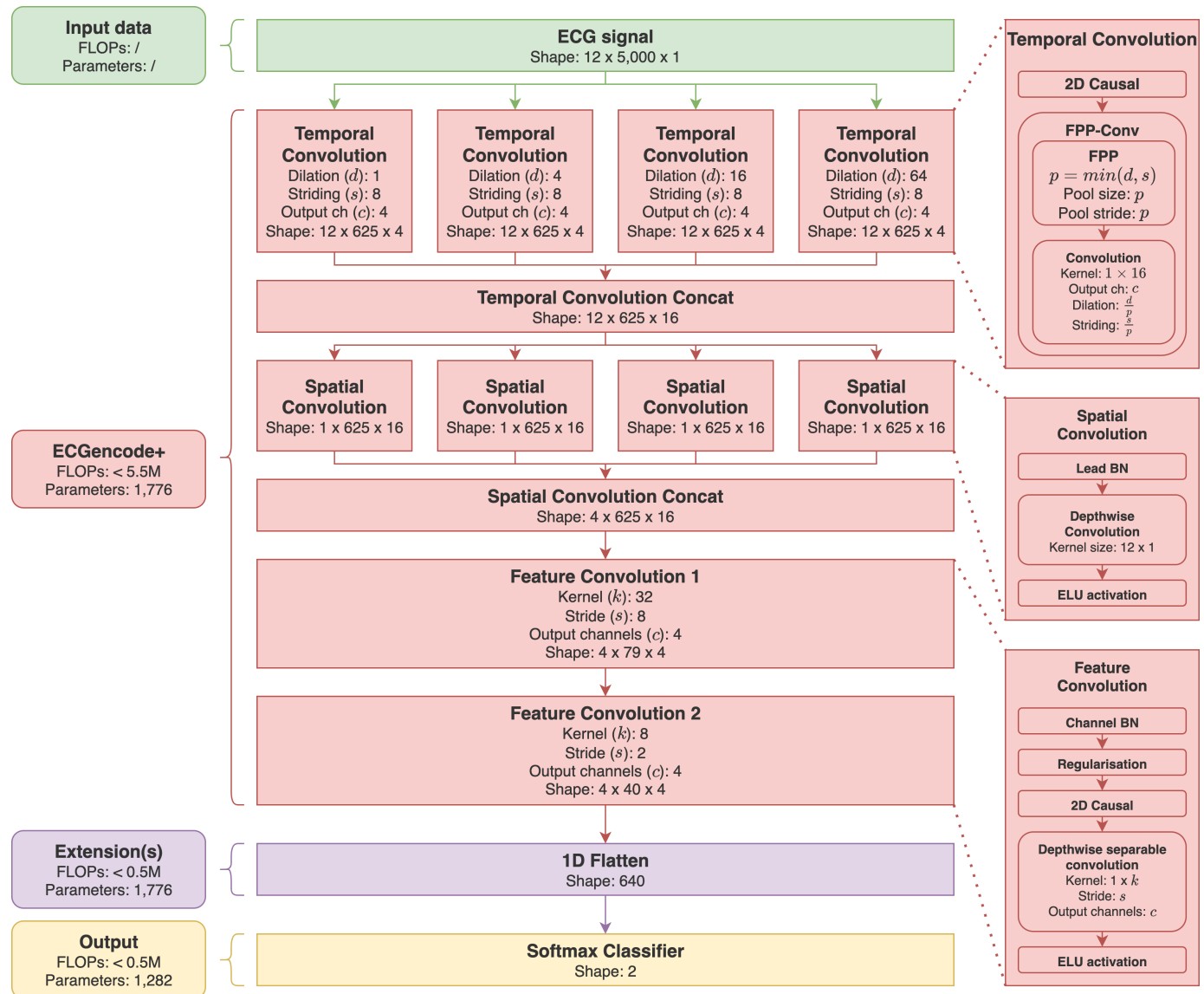

Fig. 4: Architectural overview of the proposed CSD-AFNet model for binary AFib classification. Building on the ECGencode model, it replaces the FLOPs-heavy variable kernel approach with the novel and efficient Feature-Preserving Pooled Convolution (FPP-Conv) module and adds 2D causal padding before each temporal-axis convolution. Regularisation combines 2D Spatial Dropout and Spatial Gaussian Noise. The full model has 3,058 parameters and approximately 5.5 million FLOPs.

striding and dilation while ensuring no features are skipped. The complete strategy is visualised in Fig. 5.

The FPP-Convs in CSD-AFNet use a fixed target stride of 8 and kernel size of 16 with variable target dilations. To further optimise computational efficiency, intermediate pooled outputs are reused. For instance, $p = 8$ pooling is achieved by applying an additional $p = 2$ pooling to the existing $p = 4$ pooling result. All FPP-Convs of the CSD-AFNet use an average pooling strategy.

## IV. EXPERIMENTAL SETUP

The CSD-AFNet model is trained and evaluated following experimental methodology introduced by the ECGencode study [3] and the authors of the PTB-XL dataset [12].

### A. Datasets

TABLE I: Definition of the evaluated AFib classification tasks. For all tasks, ECGs from AFib-negative patients are used as negative samples.

| Task | Positive samples |
| --- | --- |
| Detection | ECGs from AFib-positive patients during an active AFib episode |
| NSR | ECGs from AFib-positive patients after a Detection sample, but without an active episode |
| Prediction | ECGs from AFib-positive patients recorded prior to the first Detection sample |
| Related | All ECGs from AFib-positive patients (i.e., Detection, NSR, and Prediction) |

The publicly available CODE-15% dataset [13] is used for

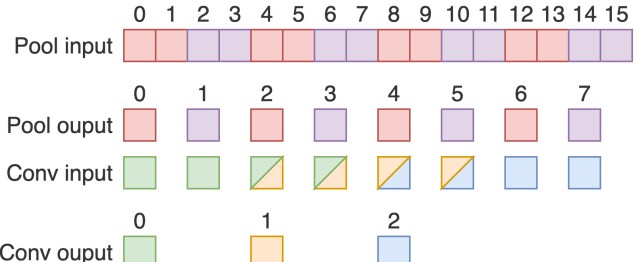

Fig. 5: Visualisation of the proposed Feature-Preserving Pooled Convolution (FPP-Conv), which avoids feature skipping when combining striding and dilation (see Fig. 2). Given target stride $s=4$ and dilation $d=2$, average pooling with $p=\min(d,s)=2$ is applied first. This yields residual stride $s_{\text{res}}=\lceil s/p \rceil=2$ and residual dilation $d_{\text{res}}=\lceil d/p \rceil=1$, enabling a regular convolution.

training and evaluation. CODE-15% consists of real-world 12-lead ECGs acquired across multiple clinical sites and acquisition devices from the Telehealth Network of Minas Gerais. A stratified ten-fold split is used to preserve the distribution of diagnostic labels, patient age and sex while ensuring no patient overlap. Folds 1-7 are used for training, 8 for validation and 9-10 for testing.

For out-of-distribution evaluation, the PTB-XL dataset [12] is used. It originates from a different clinical cohort in a different geographical location, featuring unseen devices and patients. The predefined folds 1-2 are used for the calibration split, folds 3-10 for the test split.

Minimal preprocessing is done, with each dataset being normalised in a global fashion using the train/calibration split and CODE-15% being upsampled to 500Hz to match the PTB-XL sampling frequency. To support reproducibility, the fold assignments used in this study can be made available upon reasonable request.

Table I shows the definition of four distinct tasks that can be derived from the different dataset splits. Table II shows the number of samples present in the data splits for these tasks. In CODE-15%, ECG order is estimated using patient age, since acquisition dates are unavailable. This may cause some Prediction samples to be labelled as NSR, making the NSR task harder.

TABLE II: Sample count in the train, validation and test split of CODE-15% and the calibration and test split of PTB-XL.

| Dataset split | Related | Detection | NSR | Prediction | Negative |
|---|---|---|---|---|---|
| C15% Train | 7,888 | 4,913 | 1,298 | 1,461 | 234,435 |
| C15% Val | 1,153 | 699 | 214 | 204 | 33,241 |
| C15% Test | 2,226 | 1,420 | 319 | 434 | 66,778 |
| PTB-XL Cal | 354 | 302 | 32 | 20 | 4,002 |
| PTB-XL Test | 1,437 | 1,212 | 141 | 84 | 16,006 |

### B. AFib Classification Models Configuration

The model configuration of CSD-AFNet is illustrated in Fig. 4. The baseline models included for comparison are the ECGencode model, the Attia model and AFibri-Net variants 3 and 5. Their hyperparameter configuration is adopted from the ECGencode study. To enable a direct and unambiguous comparison of computational cost, the CSD-AFNet hyperparameters are chosen to mirror the intermediate output shapes and temporal coverage of the ECGencode model.

All models are trained for 1000 epochs on the CODE-15% train set using the Related task labels. This unified strategy addresses class imbalance and feature overlap between subtasks. The model checkpoint with highest validation sensitivity is used for evaluation. Training uses a batch size of 128, AdamW optimiser, and categorical focal loss. To validate the unified strategy, CSD-AFNet is also fine-tuned for 200 additional epochs using task-specific labels and the same selection criterion.

### C. Evaluation Metrics

The reported metrics include the the area under the ROC curve (AUC) as well as the threshold-dependent metrics F1, sensitivity, specificity, and precision. Used model thresholds are optimised per evaluation task by maximising validation or calibration F1. Metrics are reported as point estimates on the test set. For AUC and F1, confidence intervals are derived from 10,000 bootstrap iterations. This supports statistical significance assessment via non-overlapping intervals, following the methods in PTB-XL [12] and ECGencode [3].

Computational efficiency is assessed by measuring the number of FLOPs (as calculated by the keras-FLOPs library[1]) and the number of trainable parameters.

## V. RESULTS AND ANALYSIS

This section evaluates the proposed CSD-AFNet model in terms of classification performance and computational efficiency compared to the ECGencode, Attia, and AFibri-Net (3 and 5) models.

### A. AFib Classification Performance

Table III summarises classification performance for the Related task and the Detection, NSR and Prediction subtasks on the CODE-15% and PTB-XL test set.

CSD-AFNet consistently achieves significantly higher AUC scores than both AFibri-Net variants across all tasks and comparable scores to the ECGencode and Attia models, which represent the current SOTA in this domain.

With respect to F1 scores, the Attia model achieves superior performance on the Detection task. This Detection task superiority also translates to a better F1 score for the related task, which is an aggregate of all subtasks with the majority of positives being detection samples as visible in Table II. Notably, even for the F1 scores, CSD-AFNet significantly surpasses all F1 scores reported by AFibri-Net models, except for AFibri-Net 5 in the Prediction task. The reported F1 scores, although low in absolute value, reflect results of leading models in other studies, with the CSD-AFNet precision of

---

[1]https://pypi.org/project/keras-FLOPs/

TABLE III: Comparative performance of the evaluated AFib classification models across four tasks and two datasets. All models were trained using the CODE-15% train split with the Related task labelling scheme, and evaluated on the test splits of CODE-15% and PTB-XL. Reported metrics include AUC, F1-score, sensitivity, specificity, and precision. Confidence intervals for AUC and F1 are derived from 10,000 bootstrap iterations on the test set. Thresholds for classification were selected to maximise F1 performance on a per-task basis using the validation/calibration split. Bold values indicate top-performing models per task. Asterisks (*) denote statistically significant differences (non-overlapping confidence intervals with the best score).

| Task | Model | AUC | F1 | Sensitivity | Specificity | Precision | PTB-XL AUC | PTB-XL F1 |
|---|---|---|---|---|---|---|---|---|
| Related | CSD-AFNet | **0.9421 ± 0.0064** | 0.6477 ± 0.0168* | 0.6438 | 0.9885 | 0.6517 | **0.9390 ± 0.0075** | 0.6717 ± 0.0187* |
| | ECGencode | 0.9385 ± 0.0063 | 0.6122 ± 0.0170* | 0.6366 | 0.9852 | 0.5897 | 0.9332 ± 0.0076 | 0.6500 ± 0.0199* |
| | Attia | 0.9365 ± 0.0066 | **0.6941 ± 0.0162** | 0.6554 | 0.9922 | 0.7376 | 0.9323 ± 0.0083 | **0.7272 ± 0.0198** |
| | AFibri-Net 3 | 0.8843 ± 0.0080* | 0.3211 ± 0.0195* | 0.2686 | 0.9865 | 0.3989 | 0.8508 ± 0.0101* | 0.4295 ± 0.0194* |
| | AFibri-Net 5 | 0.9110 ± 0.0078* | 0.6046 ± 0.0163* | 0.6725 | 0.9816 | 0.5492 | 0.9106 ± 0.0096* | 0.7224 ± 0.0201 |
| Detection | CSD-AFNet | 0.9883 ± 0.0023 | 0.7097 ± 0.0185* | 0.7507 | 0.9922 | 0.6730 | 0.9719 ± 0.0043* | 0.7043 ± 0.0194* |
| | CSD-Finetuned | **0.9895 ± 0.0022** | 0.7435 ± 0.0176* | 0.7725 | 0.9935 | 0.7165 | **0.9795 ± 0.0032** | 0.7414 ± 0.0185* |
| | ECGencode | 0.9851 ± 0.0024 | 0.6691 ± 0.0186* | 0.7592 | 0.9892 | 0.5982 | 0.9653 ± 0.0049* | 0.6882 ± 0.0208* |
| | Attia | 0.9853 ± 0.0036 | **0.8042 ± 0.0160** | 0.8275 | 0.9951 | 0.7823 | 0.9750 ± 0.0046 | **0.7873 ± 0.0183** |
| | AFibri-Net 3 | 0.9308 ± 0.0064* | 0.3308 ± 0.0241* | 0.2965 | 0.9895 | 0.3742 | 0.8764 ± 0.0097* | 0.4432 ± 0.0229* |
| | AFibri-Net 5 | 0.9747 ± 0.0045* | 0.6335 ± 0.0178* | 0.8648 | 0.9816 | 0.4998 | 0.9626 ± 0.0070* | 0.7799 ± 0.0193 |
| NSR | CSD-AFNet | **0.9032 ± 0.0200** | 0.2587 ± 0.0388 | 0.3480 | 0.9936 | 0.2059 | **0.7770 ± 0.0390** | 0.0699 ± 0.0236 |
| | CSD-Finetuned | 0.8798 ± 0.0218 | 0.1599 ± 0.0310* | 0.2476 | 0.9912 | 0.1181 | 0.7703 ± 0.0389 | 0.0905 ± 0.0316 |
| | ECGencode | 0.8949 ± 0.0201 | 0.2158 ± 0.0349 | 0.3292 | 0.9918 | 0.1606 | 0.7663 ± 0.0375 | 0.0766 ± 0.0484 |
| | Attia | 0.8831 ± 0.0205 | **0.2776 ± 0.0422** | 0.3386 | 0.9947 | 0.2353 | 0.7000 ± 0.0421 | 0.0347 ± 0.0418 |
| | AFibri-Net 3 | 0.8278 ± 0.0253* | 0.0991 ± 0.0253* | 0.1567 | 0.9904 | 0.0725 | 0.7133 ± 0.0433 | 0.0220 ± 0.0340 |
| | AFibri-Net 5 | 0.8346 ± 0.0265* | 0.1724 ± 0.0242* | 0.4577 | 0.9816 | 0.1062 | 0.6346 ± 0.0406* | **0.0993 ± 0.0488** |
| Prediction | CSD-AFNet | 0.8224 ± 0.0233 | 0.1402 ± 0.0283 | 0.1728 | 0.9916 | 0.1179 | 0.7352 ± 0.0613 | 0.0430 ± 0.0448 |
| | CSD-Finetuned | 0.8196 ± 0.0232 | 0.0988 ± 0.0239* | 0.1336 | 0.9898 | 0.0784 | 0.7295 ± 0.0598 | 0.0259 ± 0.0315 |
| | ECGencode | **0.8235 ± 0.0222** | 0.1047 ± 0.0231 | 0.1590 | 0.9878 | 0.0781 | **0.7491 ± 0.0547** | **0.0690 ± 0.0609** |
| | Attia | 0.8226 ± 0.0216 | **0.1597 ± 0.0323** | 0.1751 | 0.9934 | 0.1467 | 0.7066 ± 0.0596 | 0.0241 ± 0.0304 |
| | AFibri-Net 3 | 0.7740 ± 0.0241* | 0.0614 ± 0.0198* | 0.0783 | 0.9904 | 0.0504 | 0.7124 ± 0.0493 | 0.0576 ± 0.0586 |
| | AFibri-Net 5 | 0.7688 ± 0.0241* | 0.1113 ± 0.0205 | 0.2258 | 0.9816 | 0.0739 | 0.6230 ± 0.0529* | 0.0664 ± 0.0465 |

0.2059 on the NSR task reflecting a number needed to screen of 5, which is also in line with leading models [1], [3], [14].

For the calibrated CSD-AFNet variant, F1 improves on the Detection task, but performance on the other subtasks worsens, showing overfitting tendencies. This supports the effectiveness of the unified training strategy in handling class imbalance and shared features across subtasks.

For out-of-distribution evaluation on PTB-XL, models are not retrained and only threshold calibration is applied. The overall performance is lower and the F1 score is less representative given that no retraining is done. Nonetheless, the relative ranking between models remains consistent with the CODE-15% results, suggesting appropriate generalisability beyond this training dataset.

Given that active AFib detection is feasible through algorithmic integration into medical edge devices already, greater emphasis is placed on the more challenging NSR and Prediction subtasks in this study. For these subtasks, CSD-AFNet demonstrates strong performance, on par with both the ECGencode and Attia models, while outperforming the AFibri-Net variants, supporting its clinical relevance.

### B. Computational Efficiency Analysis

Table IV provides a comparative overview of computational efficiency across the evaluated binary classification models. CSD-AFNet exhibits the lowest FLOPs and parameter count among all evaluated architectures. Specifically, it more than halves the parameter count and achieves a 15-fold reduction in FLOPs compared to the ECGencode model, while maintaining comparable classification performance. Compared to the the

TABLE IV: Comparison of computational efficiency across the evaluated AFib classification models. Parameter counts and FLOPs are reported, along with their proportions relative to the proposed CSD-AFNet model (baseline: 1×).

| Model | Parameters | | FLOPs | |
|---|---|---|---|---|
| CSD-AFNet | **3,058** | (1x) | **± 5.5M** | (1x) |
| ECGencode | 8,242 | (2.5x) | ± 83.5M | (15x) |
| Attia | 217,350 | (71x) | ± 670M | (122x) |
| AFibri-Net 3 | 191,106 | (62.5x) | ± 8.5M | (1.5x) |
| AFibri-Net 5 | 366,594 | (120x) | ± 15.5M | (3x) |

Attia model, it has a significant 71x reduction in parameters and 122x reduction in FLOPs.

## VI. DISCUSSION AND CONCLUSIONS

This study introduced the CSD-AFNet model which uses novel FPP-Convs and ECG-specific 2D causal padding, to deliver classification performance on par with SOTA methods for various AFib classification tasks with significantly improved computational efficiency.

### A. Summary of Findings

With only 3,058 parameters and approximately 5.5M FLOPs, CSD-AFNet demonstrates remarkable computational efficiency. Compared to the benchmark Attia model [1], it achieves a 71-fold reduction in parameters and a 122-fold reduction in FLOPs. Relative to the ECGencode model, which serves as its architectural basis, CSD-AFNet reduces the parameter count by more than half and lowers the FLOPs count by approximately 15-fold.

These reductions address a key limitation of the ECGencode model, which, although computationally efficient in terms of parameters, retained a higher FLOPs count than AFibri-Net 3. CSD-AFNet surpasses AFibri-Net 3 across all evaluated tasks, offering significantly higher AUC and F1 scores while using 62.5 times fewer parameters and 1.5 times fewer FLOPs.

## B. Practical and Clinical Implications

For the benchmarked AFibri-Net 3 model, the feasibility of inference on resource-constrained hardware has already been demonstrated [5]. Given its substantially lower FLOPs and parameter count with significantly better classification performance that matches SOTA models, CSD-AFNet offers an even more viable solution for real-time deployment on medical edge devices, including integration within ECG acquisition systems.

In training scenarios, models with fewer parameters are less demanding in terms of required GPU resources, resulting in overall lower operational costs. This is especially advantageous for smaller clinical centres or institutions with limited budget that aim to train models locally on their own datasets with limited computational resources.

Furthermore, CSD-AFNet's efficiency aligns well with the architectural constraints of emerging hardware platforms, such as photonics-based neural accelerators. Its minimal parameter and FLOPs counts facilitate integration with such fixed-structure systems, supporting scalable deployment for both prospective monitoring and retrospective ECG analysis.

## C. Limitations and Future Directions

While CSD-AFNet delivers high AUC scores across all tasks and datasets with a remarkably small computational footprint, several directions of future research can further strengthen and extend this work. First, an ablation study could isolate the impact of key architectural contributions, FPP-Convs and 2D causal padding, on classification performance, informing future model designs. Given the consistently strong performance across tasks, a stratified 10-fold cross-validation with hyperparameter tuning and other optimisations may reveal classification superiority, especially on specific subtasks. Future work may also explore adapting the model to single-lead input by removing the spatial convolutions, enabling use on wearable ECG devices. Moreover, the architecture may generalise beyond AFib detection to other cardiac or physiological conditions. In terms of deployment, benchmarking inference time and energy use on real medical edge hardware would validate practical viability, with the PTB-XL experiments suggesting that pretraining on a broad cohort followed by local finetuning may be needed to optimise accuracy for specific devices and populations. Finally, releasing such pretrained CSD-AFNet weights could facilitate transfer learning for low-resource clinical institutions.

## Acknowledgment

The authors used OpenAI's ChatGPT tool to enhance readability and language during manuscript preparation. All content was subsequently reviewed and edited, and the authors take full responsibility for the final version.

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
