# OpenReview forum: "CSD-AFNet: Computationally Efficient Atrial Fibrillation Classification from ECGs using 2D Causal Strided Dilated Convolutions"
_IEEE.org/EMBS/BHI/2025/Conference — BHI 2025_

### Official Review · Reviewer_zToj · 2025-06-25
**CSD-AFNet: A Computationally Efficient Deep Learning Model for ECG-Based Atrial Fibrillation Classification using 2D Causal Strided Dilated Convolutions**

**Confidence:** 4
**Clarity Of Writing:** great
**Clinical Significance:** great
**Methodological Novelty:** great
**Overall Rating:** 7

**Experiments And Results:**

good

**Questions For The Authors:**

1. Does average pooling ever blur sharp QRS complexes?
2. Have you evaluated or at least inferred performance on PTB‑XL or Chapman‑Shaoxing?
3. Can you share milliseconds‑per‑ECG on a representative MCU or smartphone CPU?
4. You optimize F1 on the validation fold for each task. Would a cost‑sensitive or clinically informed threshold (e.g., maximize sensitivity at 95 % specificity) change conclusions?

**Strengths:**

1. A 71× parameter and 122× FLOP reduction relative to a strong ResNet baseline is impressive and well‑quantified (Table III).
2. FPP‑Conv is an elegant workaround for the well‑known “skipped feature” problem when combining stride and dilation; the explanatory figures (pp. 3‑5) are clear.
3. The paper separates “Detection,” “NSR,” and “Prediction” scenarios (Table I) and discuss why NSR & Prediction matter most in practice (Sec. V‑A).
4. The manuscript is well structured, and graphics effectively convey architecture and results.

**Summary Of The Paper:**

The paper introduces CSD‑AFNet, a lightweight convolutional network for classifying atrial fibrillation (AFib) from 12‑lead ECGs. Its key design choices are (i) 2‑D causal padding that prevents temporal information “leakage,” and (ii) a novel Feature‑Preserving Pooled Convolution (FPP‑Conv) module that combines dilation and stride without dropping input samples. Evaluated on the public CODE‑15 % dataset, CSD‑AFNet matches or slightly exceeds the AUC of larger state‑of‑the‑art (SOTA) models while reducing computational cost. The paper showed that this efficiency enables training in resource‑constrained labs and deployment on wearables or edge devices.

**Weaknesses:**

+ Because results come only from the CODE-15 % dataset, it is unclear whether CSD-AFNet will perform equally well on ECGs collected with different hardware, sampling rates, or patient demographics. Testing on an external set such as PTB-XL, or at least discussing expected domain-shift effects, would make the conclusions more convincing.
+ Training a single “AF-related vs. control” model and then scoring three sub-tasks may under-optimize Detection, NSR, and Prediction individually. A brief fine-tune per sub-task, or evidence that fine-tuning offers no benefit, would clarify whether the current choice sacrifices performance.
+ Claims of edge-device suitability rely on theoretical FLOP savings. The paper reports no inference-time or power measurements on actual hardware. One benchmark on a low-power board would translate those savings into practical deployment terms.
+ The model’s strengths are attributed jointly to 2-D causal padding and the new FPP-Conv layer, yet no experiment separates their contributions. A simple ablation (e.g., removing or replacing each component) would show which innovation drives performance and guide future adopters.

---

### Official Review · Reviewer_Wmiz · 2025-07-07
**CSD-AFNet: A Computationally Efficient Deep Learning Model for ECG-Based Atrial Fibrillation Classification using 2D Causal Strided Dilated Convolutions**

**Confidence:** 4
**Clarity Of Writing:** good
**Clinical Significance:** excellent
**Methodological Novelty:** good
**Overall Rating:** 6
**Final Rating:** 7

**Experiments And Results:**

fair

**Questions For The Authors:**

Could you provide an ablation study isolating the contributions of FPP-Convs and 2D causal padding?
This would clarify whether performance gains primarily from one component or from their combination. Understanding this distinction would help others adapt or extend the architecture in future work.

Have you explored possible reasons for the consistently low F1 scores in the NSR and Prediction subtasks (≤ 0.26)?
Are there clinical or data-related factors such as class imbalance, label noise, or signal ambiguity that might explain this? Insights here could clarify whether performance ceilings are due to model limitations or dataset characteristics. Could fine-tuning per subtask instead of a single binary classifier improve F1 on NSR/Prediction?

Could you expand on the ECG signal pre-processing steps (e.g., resampling, filtering, segment selection, normalization)?
These implementation details are essential for reproducibility and for assessing whether the model can generalize across datasets with different acquisition conditions.

**Strengths:**

The motivation and the clinical relevance of the work is clear. As the work targets deployment in edge or resource-limited clinical settings, it has a high practical use case.

FPP-Convs solve the problem of combined striding and dilation. Extension of causal convolutions to 2D ECG signal representations is well motivated and clearly explained.

Benchmarked against multiple competitive models (Attia, AFibri-Net, ECGencode).

**Summary Of The Paper:**

This paper introduces CSD-AFNet, a computationally efficient deep learning model for atrial fibrillation (AFib) classification from ECG signals. Motivated by the high computational cost of many state-of-the-art (SOTA) AFib detection models, the work proposes Feature-Preserving Pooled Convolutions (FPP-Convs) and 2D causal convolutions. These are designed to retain full temporal context without future leakage, while reducing the model’s parameter count and floating-point operations (FLOPs). The model is evaluated on the public CODE-15% dataset across four AFib-related tasks and demonstrates comparable performance to more complex models with 71 times fewer parameters and 122 times fewer FLOPs.

**Weaknesses:**

While the impact of the full architecture is validated, there is no ablation to isolate the contribution of FPP-Convs versus 2D causal padding.

Although it is acknowledged that NSR and Prediction subtasks are more challenging, the low F1 scores (≤ 0.26) across all models, including CSD-AFNet, deserve a deeper clinical and technical reflection.

Adding dataset pre-processing details would improve reproducibility.

Please add the table captions above the table following the template.

---

### Official Review · Reviewer_o5Z1 · 2025-07-13
**CSD-AFNet: A Computationally Efficient Deep Learning Model for ECG-Based Atrial Fibrillation Classification using 2D Causal Strided Dilated Convolutions**

**Confidence:** 5
**Clarity Of Writing:** great
**Clinical Significance:** great
**Methodological Novelty:** great
**Overall Rating:** 8

**Experiments And Results:**

great

**Questions For The Authors:**

1. Why did you choose a unified binary classification model for all subtasks instead of training specialized models for Detection, NSR, and Prediction separately?

2. Have you evaluated the model’s real-time inference latency and energy consumption on actual medical edge devices or wearables?

3. What are the limitations, if any, of the FPPConvs in terms of temporal resolution or feature representation?

4. How does the model handle noisy or poor-quality ECG recordings?

**Strengths:**

1. CSD-AFNet significantly reduces parameter count and FLOPs—by up to 71x and 122× respectively which makes it ideal for low-resource and real-time deployment scenarios.

2. Despite its compact design, the model achieves classification performance comparable to state-of-the-art methods across multiple AFib classification tasks.

3. The model is designed with real-world clinical use in mind, particularly for edge devices and low-cost setups, improving accessibility and scalability.

4. Evaluation across multiple tasks (Detection, NSR, Prediction) using a public dataset, with well-established metrics and statistical significance testing, supports the robustness of the findings.

5. Diagrams and visualizations are clear, informative, and effectively support the technical descriptions.

**Summary Of The Paper:**

The paper proposes CSD-AFNet, a lightweight deep learning model for ECG-based AFib classification. It introduces FPP-Convs to combine striding and dilation without losing input features. The model also uses 2D causal padding to avoid future leakage in temporal data. CSD-AFNet achieves competitive performance to state-of-the-art models while significantly reducing parameters and FLOPs. It is evaluated on the CODE-15% dataset and is well-suited for deployment on low-resource devices.

**Weaknesses:**

The paper is methodologically sound, clearly written, and well-motivated. While there may be areas for future exploration like
- exploring fine-tuning strategies for each AFib subtask separately. This could potentially enhance F1 scores, particularly for subtasks like Prediction and NSR.
- The method is promising for AFib classification, but it would be valuable to evaluate its generalizability to other ECG classification tasks to further establish its versatility.
But these do not undermine from the strength and completeness of the current work.

---

### Official Review · Reviewer_v27v · 2025-07-14
**CSD-AFNet: A Computationally Efficient Deep Learning Model for ECG-Based Atrial Fibrillation Classification using 2D Causal Strided Dilated Convolutions**

**Confidence:** 4
**Clarity Of Writing:** good
**Clinical Significance:** fair
**Methodological Novelty:** good
**Overall Rating:** 5

**Experiments And Results:**

great

**Questions For The Authors:**

1) How do you extend the generalizability of the current approach to different devices and data being collected under different conditions (eg.: placement of device, single lead vs multiple leads)?
2) what is the difference between a 'medical device',  'medical edge device' and a wearable that authors have used in the current manuscript?
3) What is the author's opinion if smaller clinics/institutions can use models trained on huge daatsets instead of training from scratch? I specifically refer to  " especially advantageous for smaller clinical centres or institutions with limited budget that aim to train models locally on their own datasets with limited computational resources". in the clinical implications section

**Strengths:**

1) The authors setup the problem through the abstract and introduction very clearly. More specifically, it is very clear from the outset what authors are proposing and the results are directed exactly towards proving their claims
2) The authors also provide a very detailed description of the experiments undertaken. This is helpful to understand the paper better
3)  The use of graphs and tables are very clear.

**Summary Of The Paper:**

The proposed work developed a a computationally efficient DL model for Afib-related ECG classification tasks that can be particularly deployed on medical edge devices and wearables. The manuscript evaluates the proposed model on a publicly available data to demonstrate its efficacy. The proposed model extends (or builds upon) a previous SOTA model 'ECGencode' that can be accurate as well as computationally less efficient.

**Weaknesses:**

1) Methodological approaches have been loosely used. (a) statistical significance has been uses superficially as beyond table II nowhere it is mentioned what particular tests were used. In a multiple comparison, it will require a hierarchical approach (eg.: ANOVA followed by other posthoc tests). (b) Results being reported are on a single split of the data. Although authors do report CI using bootstrapping and the evidence for taking such approach is a reference paper. However, the approach used in this work is not appopriate. Ideally, the authors should have undertaken a 10-fold stratified cross-validation approach, reported mean metrics and CI based on the range. This is the standard practice
2) Some terminology like 'medical edge devices' and 'wearables' has been loosely used. For example: Medical devices are mainly 12-lead ECG whereas wearables are a single-lead ECG. From the figure, it seems the authors are using 12-lead ECG as the input.
3) The authors do not provide much details into the data, why only this data, its class imbalance and why is such a comparison ideal in the field?

---

### Official Review · Reviewer_fiqQ · 2025-07-21

[review text omitted: it was posted to a different submission]